# Creating the next Digital Telemedicine Tool for Parkinson's Disease Management with AI

Rita Duarte Vieira
*C-BER, INESC TEC*
Porto, Portugal
rita.d.vieira@inesctec.pt

Adriana Arrais
*C-BER, INESC TEC*
Porto, Portugal
adriana.g.arrais@inesctec.pt

Duarte Dias
*C-BER, INESC TEC*
Porto, Portugal
duarte.f.dias@inesctec.pt

Carolina Soares
*Department of Neurology, CHUSJ, and*
*Department of Clinical Neurosciences and Mental Health, FMUP*
Porto, Portugal
carolina.soares@chsj.min-saude.pt

João Massano
*Department of Neurology, CHUSJ, and*
*Department of Clinical Neurosciences and Mental Health, FMUP*
Porto, Portugal
joao.massano@chsj.min-saude.pt

João Paulo Silva Cunha*
*Senior Member, IEEE*
*INESC TEC, and Faculty of Engineering, University of Porto*
Porto, Portugal
jcunha@ieee.org

*Abstract*—Parkinson's Disease (PD) is a neurological disease that progresses over time and causes severe motor symptoms. Therefore, treating PD requires constant patient monitoring, which may turn clinical practice overwhelming, preventing its practical implementation, and raising the need for patient monitoring outside the clinical setting. The iHandU system described in this paper fulfils this need by providing an objective way to quantify motor symptoms of PD in non-clinical settings. It integrates an innovative real-time assessment of the severity of motor symptoms based on signal processing and Machine Learning models that mimic the clinical severity classification scales used in practice and allows for a more continuous and personalized therapy planning and management by doctors, through the use of a web dashboard user-friendly interface. This system, recently tested at 5 patients' homes, has shown promising results as a PD patient management digital platform, reaching a usability score of 83.9% (A grade) based on the System Usability Scale (SUS). Such a level shows a strong alignment between user needs, expectations and functionalities. This study highlights the potential of the used system as a Patient Management Tool showing a case study from an ongoing clinical study. By giving additional information to the doctors with features beyond the semi-quantitative rating scales currently used, allowing a more optimized and continuous PD symptom management, it will be possible to advance PD management further.

*Index Terms*—Parkinson's disease, Objective Assessment, mHealth, Clinical Study, Features Extraction, MDS-UPDRS Severity Classification, Machine Learning.

## I. Introduction

**P**ARKINSON'S Disease (PD) is a neurodegenerative disorder with a variety of symptoms, both motor and non-motor, that affect patients' quality of life. Nowadays, there is no cure but there are some therapy options that aim to relieve symptoms, such as medication and Deep Brain Stimulation (DBS). The disease's prevalence is increasing, affecting around 10 million people worldwide, with higher risk connected with age, gender, and hereditary variables, presenting a huge medical and societal issue. Clinical examinations utilize several rating scales to assess PD and examine motor symptoms and their impact on patient's lives [1], [2]. The existing methods for assessing PD motor symptoms rely on widely used medical quantitative rating scales characterized by a certain degree of intra and inter-rater variability, whilst patient-reported accounts of their condition might be erroneous and biased. Furthermore, the fact that there are few medical appointments, with a lack of medical team time, the difficulty for patients to visit the hospital regularly and that motor symptoms are only assessed in a clinical setting, prevents a more consistent and continuous disease follow-up. Objective measuring procedures for each symptom in real-time can improve the medical assessment and lead to more accurate therapy decisions for each patient.

The Biomedical Research and Innovation (BRAIN) Laboratory at the Institute for Systems and Computer Engineering, Technology and Science (INESC TEC) is developing the iHandU System, a system for real-time objective quantification of PD motor symptoms to enable at-home patient monitoring and support medical assessments during clinical appointments. With the Centro Hospitalar Universitario de São João (CHUSJ) partnership and a clinical study approved by its Ethics Committee (Ref. Number: 01/2022), it was possible to implement and test this system in a clinical setting and afterwards, move it to the patient's home with a well-defined protocol established to define how patients' monitoring would be performed at home. This paper will focus on the system description and features developed based on clinical and patient feedback, the description of the algorithms and quantitative metrics based on clinical data, and the demonstration of the system's capabilities and usability for PD motor symptoms assessment outside the clinical environment, with a case study focused on the Open/Close Hands tests.

## II. State of the Art

Self-care for PwP involves identifying and responding to changes in symptoms. This proactive strategy needs expert symptom monitoring and effective management approaches.

This work is financed by National Funds through the Portuguese funding agency, FCT - Fundação para a Ciência e a Tecnologia, within project LA/P/0063/2020. DOI 10.54499/LA/P/0063/2020.
* Corresponding Author

Traditional therapies for patients with PD have included in-person rehabilitation therapy, and cognitive behavioural testing to improve self-care. To effectively manage this disease, it is important to regularly check symptoms and adhere to therapy guidelines [3]. In recent years, the rapid development of mobile and wearable devices has enabled the integration of Internet of Things (IoT) technology into healthcare, resulting in over 100,000 available apps for PD [4]. The available mHealth technologies are improving healthcare monitoring and management, particularly for individuals with PD who need personalized care outside of traditional clinical settings [3]. Smartphones have enabled the development and use of several health apps. IoT and Artificial intelligence (AI) technologies can have a significant impact on PD. Machine Learning (ML) models trained on sensory input data can help assess motor symptoms, quantify the MDS-UPDRS severity rating scale, and identify significant fluctuations in symptoms in real-time [5]. Remote monitoring using wireless sensors and wearables can benefit PD, as shown with other diseases and only with a selection of wearables, allowing continuous care without risk of exposure [5]. Using a smartphone makes tracking symptoms possible and allows for regular monitoring over time, especially for PD patients who require intelligent care outside of clinical settings. This overcomes the limitations of standard clinical exams, which only provide a fragment of a patient's disease [3].

People with Parkinson's disease (PwP) seek to use technology for different purposes, such as communicating with medical teams, making informed decisions, and accessing correct information and social support. Accessing accurate health information online remains a priority, but concerns about the range and quality of available resources highlight the challenge of distinguishing between optimism and hype. Wearable sensors and self-tracking technologies have shown promise in empowering persons with PD, but there is still a large gap between current digital solutions and the needs of this population [6]. Despite increased collaboration among researchers, doctors, and PwP, their participation remains limited, probably due to doctors' concerns about how they interact with technology. When developing digital health solutions for PD, PwP should be an equal partner between researchers and doctors. This transformation requires a joint effort to bridge the gap between existing technological advancements and PwP's needs, paving the way for a future in which digital health in PD is truly inclusive [6], [7].

The existing landscape dedicated to mHealth applications especially designed for PD patients is growing significantly. Some applications are already being used commercially, and others are being researched. Focusing on the use of wearable sensors to monitor motor symptoms, Rey et al. [8] and Estévez-Martín et al. [9] found that while many apps focus on symptom assessment and information sharing, few provide both motor symptoms assessment and therapy optimization. Several apps, like mPower [10], PD Dr [11] and ParkNosis [12], use accelerometers and smartwatches to assess motor symptoms such as tremors, movement, and balance. Others,

such as STOP app [13], StrivePD [14], PD_Manager [15], and Fox Wearable Companion [16], focus on medication registration and management, providing detailed reports and direct communication with doctors. The PDapp [17] is a mobile-web system for PwP to self-track symptoms and connect with clinicians remotely, with an application's wearable sensor. These apps aim to improve symptom management and continuous monitoring of PwP outside of a clinical setting, using the inertial sensors incorporated in the smartphones or a wearable device connected to the smartphone, such as Apple Watch [18], Personal KinetiGraph and Kinetic 360 [19]. The use of inertial sensor technologies enables the exact evaluation of motor aspects, assessment of symptom localization, and monitoring of therapy responses, delivering useful insights beyond traditional clinical scales via ML algorithms trained with sensor data obtained in a variety of settings [20], [21]. However, the presented solutions perform continuous monitoring throughout the day, which can be challenging and may raise several problems in patient follow-up.

The objective assessment of PD motor symptoms in these technologies entails assessing symptom severity, with signal processing and ML approaches increasing assessment accuracy. Pre-processing approaches for bradykinesia assessment focus on noise reduction and signal segmentation, extracting features including amplitude, speed, hesitation, and energy-related metrics for severity categorization [22]. Kim et al. (2018) [23] use Support-Vector Machine (SVM), Decision Tree (DT) and Random Forest (RF) to estimate the motor symptoms severity using a wrist-worn device. Combining feature selection approaches with ML algorithms enhances computing efficiency and accuracy, while deep feature analysis yields delicate insights into disease progression and early warning indications. Butt et al. (2020) [24] use SVM and RF to estimate the severity of Bradykinesia symptoms, with an ANFIS correlation of $r = 0.814$. Borzì et al. (2020) [25] used Butterworth filters, Fast Fourier Transform, feature selection, DT, k-Nearest Neighbors (kNN) and SVM to predict the intensity of motor symptoms with good performance (accuracy (Acc) of 77.7%). In previous work developed in the BRAIN-Lab group [26], a Finite Impulse Response filter (FIR) for raw signal processing is employed, using Polynomial Regression (PR), kNN, and SVM to evaluate the severity of Bradykinesia, through the MDS-UPDRS score with an accuracy ranging from 50% to 60%, depending on the model.

Despite the high number of mHealth apps and increasing collaboration between researchers, doctors and patients, there is still a gap between the existing solutions and the user needs, highlighting the relevance of developing user-centered mHealth solutions to truly address the challenges patients face. A digital tool for PD management was created, with iterative feedback from doctors and patients, to address the need for objective and punctual continuous monitoring of PwP outside of the hospital, using ML models for symptoms quantification created with data collected in the hospital and tested in a home-based assessment.

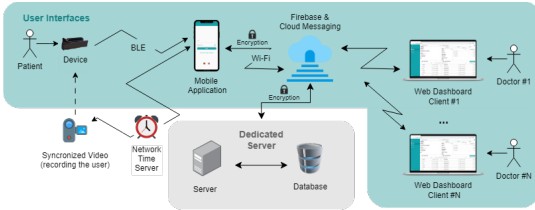

Fig. 1.  Architecture of the iHandU System.

## III. THE DIGITAL TOOL DEVELOPED FOR PD MANAGEMENT - iHANDU SYSTEM

The iHandU system, being developed by INESC TEC's BRAIN group since 2014, has seen significant improvements in hardware, signal processing, and mobile applications. The most recent version, iHandU 4.0, includes a customizable textile band with embedded sensors [27], [28], a mobile app with three data collection modes, a web dashboard for clinical monitoring, and a web server for data storage and data sharing management [17]. The iHandU device includes a SnapKi, a Bluetooth-enabled Inertial Motor Unit (IMU) and a microcontroller to collect and transmit sensor data to a smartphone with 50 Hz of sampling frequency [28]. The system is currently used in DBS surgeries [17] and CHUSJ medical appointments [26], allowing patients to quantify symptoms and manage medications as doctors monitor their disease evolution [27]. Fig. 1 shows the system's architecture, including Bluetooth Low Energy (BLE) communication, secure data storage, and notification functions using Google Firebase. To ensure patients' data security are followed the OWASP recommendations, such as the use of HTTPS encryption and login (user/password) access control is done with refresh JWT tokens. The system facilitates the assessment of PD motor symptoms with the following key functionalities:

- Using the iHandU device to monitor Parkinson's symptoms in real-time, both outside and inside clinical settings.
- Facilitating remote data sharing between doctors and patients, using the mobile application and web dashboard.
- Offering clear, comprehensive views of a patient's therapy history, including prescriptions, events that have been reported, and test results.
- Managing and prescribing medication and tests to assess symptoms severity.
- Assessing qualitative patient data, through Quality of Life Questionnaires following the PD Questionnaire-8 (PDQ-8).

A clinical study is ongoing in the hospital that comprises two phases: a clinical phase that started 2 years ago, and a new phase that has been running in the past few months to deploy such technology at patients' homes. Some changes were made to the system to implement all the requirements for this ambulatory new phase, collected from the doctors and patients before starting the clinical study at home. In this paper, the final version used by the patients at home, to test the system, is presented (Fig. 1).

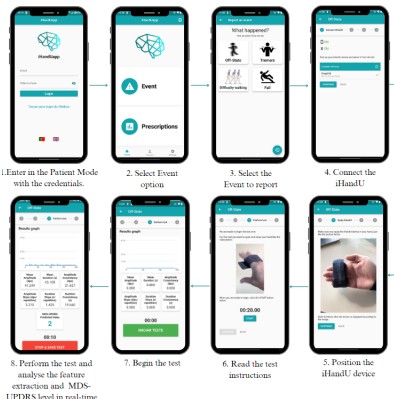

Fig. 2.  Mobile Application flow in Patient Mode to perform a test, extracting the features and the MDS-UPDRS label in real-time.

### A. iHandU Mobile Application

The iHandU mobile application, developed in Flutter using the Dart language, represents a novel approach to PD symptom assessment, facilitating remote measure of motor symptoms through signal processing and Machine Learning classification models [29]. This mobile application is based on real-world usability feedback from patients and doctors, which has been tested and will be further explained in this paper.

*1) Clinical Mode:* The Clinical Mode is designed to collect clinical data from patients without the need for an internet connection or network to focus its use on collecting raw data correctly, as well as its medical label for later classification. It allows researchers to access and download structured raw data that has not been sent to the Web Dashboard and that is stored locally in *.csv* files, facilitating researchers to quickly analyze data and improve models.

*2) Patient Mode:* The patient mode allows users (patients) to manage their condition, including reporting events, doing tests prescribed by doctors, associating events, managing medication and QoL questionnaires. The tests used to measure motor symptoms severity were designed according to Section 3 from the MDS-UPDRS items, addressing bradykinesia, tremor, gait and rigidity. When the patient feels their health condition changing, a test should be made, which will for sure complement the other tests requested by the doctors - this type of test is defined as an event. When performing a test, associated with an event or that has been prescribed, the patient must connect and position the iHandU device as directed. During the test, several real-time features are extracted together with the corresponding MDS-UPDRS label, predicted using ML models. These features are shown in the interface, as presented in Fig. 2. The ML models were trained with the MDS-UPDRS clinical label and then were implemented in the smartphone for real-time processing and classification. After completing the test, users can provide additional information that is sent with the test results to the system database.

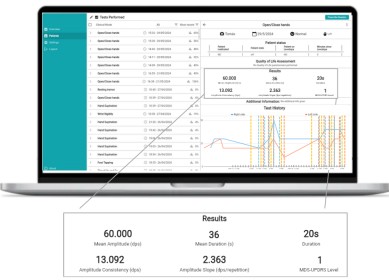

Fig. 3. Test page of the Web Dashboard: on the left, the list of tests carried out by the patient is displayed; by selecting a test, the individual view of the test is displayed on the right, with the QoL and PDQ-8, and timeline with the results of the various tests and intake medication.

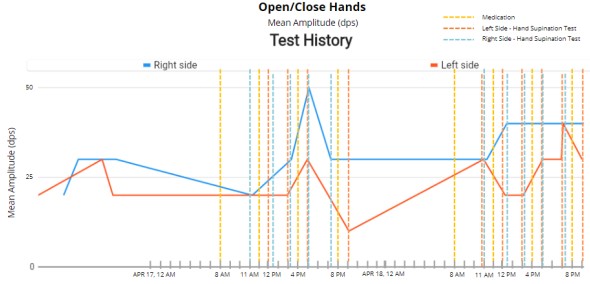

Fig. 4. Test result history displayed on the Web Dashboard, indicating medication time for additional information.

### B. Web Dashboard

The Web Dashboard is a tool (also developed in Flutter) for doctors to manage their patients, by performing follow-up analysis and prescriptions (medication intake and tests). The patient profile provides a thorough page of the patient's details, including personal data, medicine prescriptions, tests completed, and events reported. The dashboard streamlines the patient registration process by showing a list with a grid view of every patient and the ability to add new patients directly through the dashboard [29]. Additionally, doctors can visualize a temporal evolution of the patient's symptoms severity over time and correlate that results with medication intake, resulting in a more personalized and effective therapy management. In the test section of the Web Dashboard, shown in Fig. 3, it is possible to access the results of the tests performed by the patients, presenting the features and respective MDS-UPDRS labels. The Web Dashboard also contains a history section, represented in Fig. 4, with the variation of the MDS-UPDRS label throughout the carried out tests, with the indication of each patient's respective medication intake time, allowing doctors to analyse the patient's symptomatic profile throughout each day, as additional information for possible adjustment of the therapy.

## IV. CLINICAL STUDY

The clinical study with CHUSJ started a few years ago and is still ongoing. The results presented in this paper aim to show the main clinical and patient home achievements related to the following objectives of the study:

TABLE I
HOSPITAL DATASET CHARACTERIZATION BY TEST AND MDS-UPDRS
SEVERITY LABEL (TOTAL = 347 SIGNALS).

| MDS-UPDRS Label | Wrist Rigidity | Open-Close Hands | Hand Supination | Resting Tremor | Postural Tremor | Kinetic Tremor | Foot Tapping | Timed Up and Go |
|---|---|---|---|---|---|---|---|---|
| 0 | 2 | 6 | 9 | 24 | 21 | 24 | 10 | 37 |
| 1 | 17 | 7 | 10 | 10 | 16 | 8 | 13 | 19 |
| 2 | 12 | 21 | 15 | 5 | 2 | 3 | 11 | 8 |
| 3 | 7 | 12 | 5 | 2 | 0 | 0 | 4 | 4 |
| 4 | 0 | 1 | 0 | 0 | 0 | 0 | 0 | 2 |
| Total | 38 | 47 | 39 | 41 | 39 | 35 | 38 | 70 |

- Collect system requirements and feedback for system final adjustments before going to patients' homes.
- At the clinical level to collect labelled data for model research and development.
- At home to test patients' engagement with the system and also to use the model provided.
- Collaborate with doctors to allow them to prescribe tests and medication, making use of all the collected data at home for better therapy decisions.

### A. Clinical Study at the Hospital

A protocol was developed to collect data from PD patients' medical appointments and Levodopa Challenge Tests (LCT), seamlessly without interfering with their normal flow, with the wearable device and the mobile application Clinical Mode. The protocol, based on the MDS-UPDRS Part III for motor symptom quantification [30], instructed to the patient by the doctor, was adapted for hospital conditions and patient disabilities. The data collection protocol involves recording the test with a camera, and doctors assigning severity labels (0 to 4) based on the MDS-UPDRS scale. All participants agreed and signed informed consent, according to the Ethics Committee approval decision (nº 01/2022), and the data was pseudonymized. The dataset resulting from all these collections is shown in Table I and is being used for algorithm development and improvement.

### B. Clinical Study at Home

The scenario of use outside the hospital environment (at patients' homes) was designed so that patients could take advantage of all the system functionalities, such as medication reminders and confirmation, the use of the feature to report events when necessary and the execution of tests prescribed by the doctor in charge. With the help of neurology specialists, a standardized methodology for gathering data was developed to test this scenario and to expand the system's application outside of the hospital, to improve patient evaluation and at-home disease care. For the study, five patients with different levels of comfort with technology were chosen (assessed by questionnaires before the study), whose demographic data is shown in Table II.

Both end users of the system, both patients and doctors, adhered to a comprehensive protocol with different levels of engagement depending on their responsibilities, presented in Fig. 5. During the 1st phase, doctors oversaw direct patient interaction with the system. Both user types actively engaged with one another via the system throughout the 2nd phase.

TABLE II
DEMOGRAPHIC DATA OF THE 5 SELECTED PATIENTS.

| Code | Doctor Code | Age | Gender | Year of Diagnosis | Comfort Level with Technologies (1=Low, 5=High) |
|---|---|---|---|---|---|
| HPD01 | D2 | 64 | Male | 2021 | 5 |
| HPD02 | D1 | 59 | Female | 2019 | 2 |
| HPD03 | D2 | 40 | Female | 2023 | 1 |
| HPD04 | D1 | 51 | Female | 2012 | 3 |
| HPD05 | D2 | 65 | Male | 2019 | 4 |

TABLE III
HOME ASSESSMENT DATASET DISTRIBUTION BY TEST AND PATIENT.

| Patient Code | Wrist Rigidity | Open-Close Hands | Hand Supination | Resting Tremor | Foot Tapping | Timed Up and Go | Total per Patient |
|---|---|---|---|---|---|---|---|
| HPD01 | 13 | 17 | 2 | 14 | 17 | 15 | 78 |
| HPD02 | 0 | 0 | 0 | 19 | 0 | 0 | 19 |
| HPD03 | 22 | 25 | 23 | 0 | 25 | 23 | 118 |
| HPD04 | 1 | 32 | 32 | 32 | 30 | 33 | 160 |
| HPD05 | 1 | 62 | 62 | 63 | 71 | 72 | 331 |
| Total per Test | 37 | 136 | 119 | 128 | 143 | 143 | 706 |

Throughout the data collection period, through questionnaires that were answered by the participants, both patients and doctors offered input on the usability of the system to pinpoint areas that needed further improvements.

*1) Data Collection Protocol for Patients:* The patient's home-based assessment procedure included utilizing the mobile application Patient Mode and the wearable device to perform tests and confirm each medication intake, three days before their next medical appointment, which was approximately one month apart. According to the protocol presented in Fig. 5, patients had to confirm each medication intake and complete certain tests before and after taking the medicine, ideally an hour before and after. This methodical approach aims to enhance disease management and comprehend the system's usefulness in day-to-day activities, as well as its incorporation into patients' daily lives.

*2) Data Collection Protocol for Doctors:* The role of the doctors during the data collection phases differs from the patients, as shown in Fig. 5. In the 1$^{st}$ phase, doctors mostly oversaw medication intakes and test results using the Web Dashboard. Following the MA2 (Fig. 5), doctors assumed a more proactive role, using the Web Dashboard to prescribe additional tests to assess system interaction and test doctor-patient communication. The role of the doctors started long before the patients, with the preparation of the protocol and the gathering of requirements to improve the system. Once the system was ready, the doctors chose 5 patients to test the system at home for 3 months. From the protocol carried out, the dataset presented in Table III was created, resulting from the tests that were carried out by the patients, without supervision and the MDS-UPDRS label given by the doctors.

During the clinical study, in addition to testing the usability of the system, the results obtained were also analyzed. In this paper, the data collected from the patients' Open/Close Hands test was used as a focus for this paper, since it has shown to be one of the most promising results, taking into account the symptom profile of the patients in the study and the predominance of this symptom in most patients.

## V. REAL TIME OPEN/CLOSE HANDS ASSESSMENT

One of the symptoms that the iHandU system provides real-time assessment is bradykinesia, with the Open/Close Hands test, by extracting features and using a trained ML model to classify the results according to the MDS-UPDRS label. To extract features and create the Machine Learning models implemented in the system, MDS-UPDRS labelled data that has been gathered in a hospital setting is used, corresponding to the Open/Close Hands clinical-based labelled dataset presented in Table I with 47 signals.

### A. Signal Processing and MDS-UPDRS Severity Classification Models

Pre-processing these raw signals is essential before they can be utilized in the models. The procedure includes the extraction of raw signal files from *.csv* files that are stored by the mobile application. To guarantee high accuracy, raw signals are examined alongside video-recorded data, and each signal's features are extracted, such as cycle mean duration, opening angle, amplitude velocity, slope, and other signal features, based on previously developed work [26]. These features create a new dataset for ML model training, along with medical labelling. Using ANOVA statistical test for normally distributed features and the Kruskal-Wallis test for those that are not, there are some features, such as *mean_duration*, *totalpower*, *mean_angle* and *amplitude_consistency*, with a statistically significant difference (p-value<0.05), whose features values differentiate more clearly between classes and others that do not, as shown in Fig. 6.

Since the imbalance of classes in the dataset is notorious, data augmentation, namely Synthetic Minority Oversampling Technique (SMOTE), was used to solve this problem, by oversampling the minority classes. However, as the class with MDS-UPDRS equal to 4 contained only 1 element, this class was excluded from the dataset to use in the model. By applying SMOTE to the training data, a big part of the models showed reduced overfitting, while the more sensible ones did not show significant improvements. The new dataset was thoroughly examined as part of the data pre-processing, which also included handling missing values, performing exploratory data analysis, and looking for feature correlations. To ensure uniform feature contribution to the model and enhance model generalization, the features were normalized. Class-weight adjustments and hyperparameter optimization were used to enhance model performance. Considering the existing literature, 5 models were tested for the MDS-UPDRS classification of Open/Close Hands signals: the k-NN, SVM, RF, DT and PR. Based on the results obtained, shown in Table IV, one of the models was chosen and implemented in the mobile application. To accomplish this implementation, the model was chosen based on the possibilities available in the library used in Flutter, Sklite [31]. The various models tested have difficulty in accurately distinguishing between particular classes despite improvements and attempts with additional features. These models especially struggle to differentiate between classes 1 and 2, and 2 and 3. This suggests the need

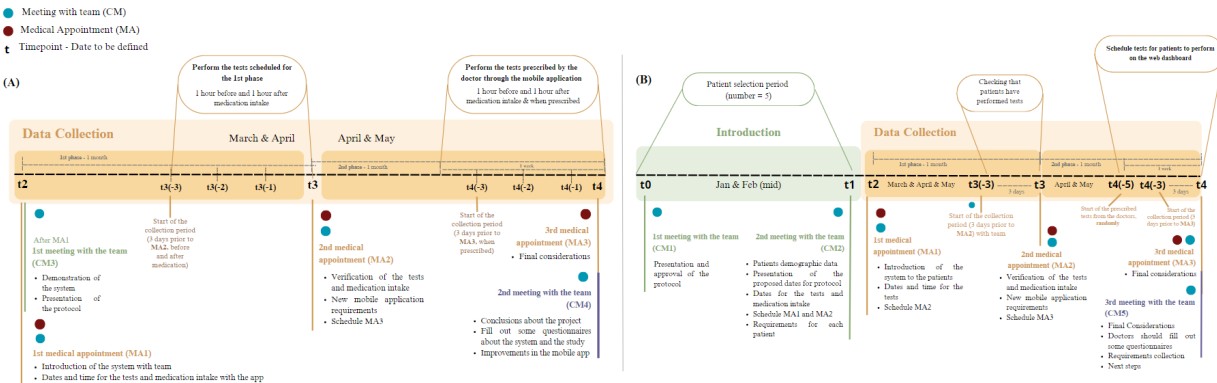

Fig. 5. **(A)**Data collection protocol for patients: In the 1st, the patients underwent the tests prescribed by the research team, according to their symptom profile, for 3 days (t3(-3), t3(-2) and t3(-1)). In the 2nd phase, beginning with MA2, the patients underwent the tests prescribed as in the 1st phase, with the addition of tests randomly prescribed by the doctors in charge, on the 3 days before the last appointment (MA3). **(B)** Data collection protocol for doctors.

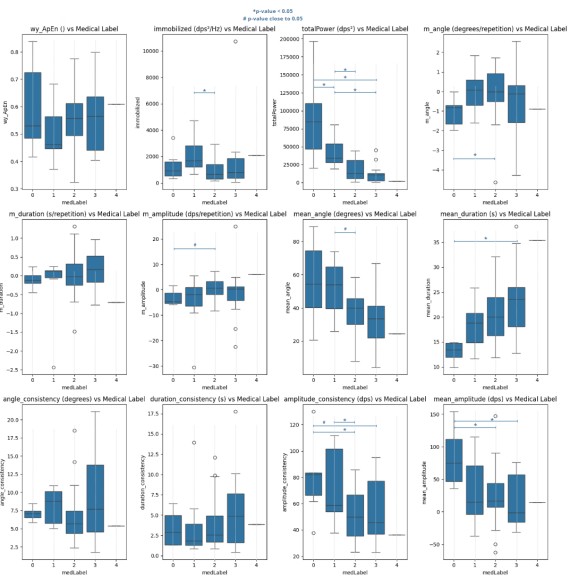

Fig. 6. Relation between the MDS-UPDRS Label and the values of the features for dataset characterisation.

for additional strategies, such as advanced feature engineering or enhanced data collection for accurate labelling, to improve model generalization and training quality.

TABLE IV
PERFORMANCE METRICS FOR DIFFERENT TESTED MODELS FOR THE OPEN/CLOSE HANDS TEST, VARYING THE NUMBER OF FEATURES, IN TRAIN AND TEST SETS.

| Models | | KNN | SVM | RF | DT | PR |
|---|---|---|---|---|---|---|
| 2 features | Train Acc | 61.11% | 58.33% | 72.22% | 83.33% | 52.77% |
| | Test Acc | 40.00% | 20.00% | 40.00% | 40.00% | 30.00% |
| 3 features | Train Acc | 63.89% | 63.89% | 69.44% | 91.67% | 63.89% |
| | Test Acc | 60.00% | 30.00% | 40.00% | 40.00% | 30.00% |
| All features | Train Acc | 55.56% | 86.11% | 100.00% | 91.67% | 100.00% |
| | Test Acc | 40.00% | 40.00% | 30.00% | 40.00% | 50.00% |

## B. Case Study

To evaluate the performance of the algorithm in a non-clinical scenario (patients' homes), a random patient was cho-sen as a case study to represent the system outcomes and the collected data. During 5 days out of the 6 days of the protocol, patient HPD05 performed 62 Open/Close Hands tests. With the recorded data, an analysis of the features extracted and the MDS-UPDRS label was performed both before and after the medication intake, to maintain a parallel analysis between the two metrics, features and label. Medication was administered at 10 am, 12 pm, and 4 pm, while the tests were performed 1h before and after medication, each day. This was prescribed by a doctor using the Web Dashboard.

The analysis of the results obtained for the 12pm medication intake in 5 days of the protocol, throughout five test sets (1 hour before and 1 hour after medication intake), show differences in several features, displayed in Fig. 7. The obser-vations indicate that the effect of the medication intake in some features has coherent results over time, as is the case with Ap-proximate Entropy (*Wy_ApEn*) and the variation in amplitude over time (*m_amplitude*), where the difference between the test sets is always negative or positive, respectively. Compared to the others, the behaviour of these two features is more in line with what would be expected, based on the existing values for each clinical MDS-UPDRS level and the significance of the features at a physiological level. However, the other features behave differently when taking medication. This different behaviour, combined with the fact that the data analyzed was acquired in an unsupervised way, does not guarantee that the improvement obtained in the *Wy_ApEn* and *m_amplitude* features is a direct consequence of the medication, and could be affected by other factors. It is therefore necessary to validate these data, taking into account the other motor symptoms and in a supervised manner to draw valid conclusions. These irregular fluctuations show the complexity of characterizing the disease, expressing inconstant effects for different features, indicating the necessity for an individualized and personalized approach to therapy adjustment.

To evaluate the effect of the medication, a statistical analysis was carried out using the Wilcoxon test [32]. The Wilcoxon test, a non-parametric test that assesses whether there are

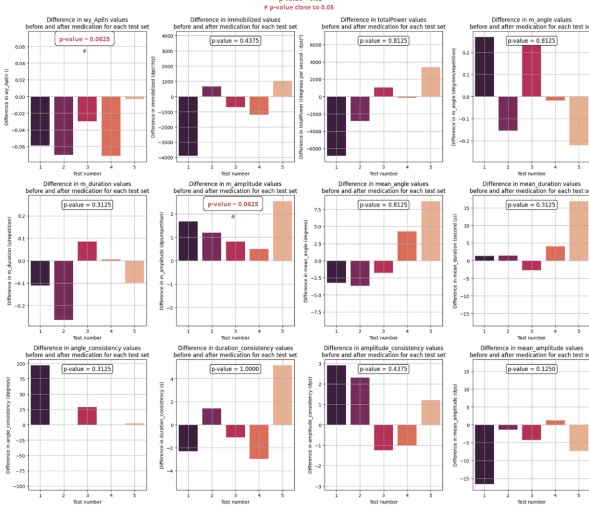

Fig. 7. Difference between the values of each feature extraction, before and after medication from 12 pm, from the Open/Close Hands test in the 5 days of the protocol at home, using data from patient HPD05.

significant differences between two sets of samples, before and after medication for each feature, was used because the sample is small and the data does not follow a normal distribution. The p-values relatively close to 0.05 indicate that there may be a tendency towards a difference, which could be further investigated with a larger sample size. The features *Wy_ApEn* and *m_amplitude*, have p-values of 0.0625, close to the common significance level of 0.05. This indicates that, at a significance level of 0.05, there is not enough evidence to reject the null hypothesis that there is no difference between the before and after measurements for any of the characteristics. However, there is a difference with *Wy_ApEn* and *m_amplitude* values, but the small sample used is not strong enough to conclude statistical significance with confidence.

After predicting the MDS-UPDRS label based on feature values, it is possible to analyse the UPDRS fluctuation throughout the day and across different medication intakes, as described in Fig. 4, provides crucial insights into therapy effects, allowing doctors to detect significant changes in symptoms patterns before and after medication administration. As seen in the instance of patient HPD05, these insights aim to help doctors keep an eye on the effectiveness of their therapies and make well-informed and personalized decisions on patient care. However, although the difference in feature values is notable when predicting the MDS-UPDRS label with *Wy_ApEn* and *m_amplitude*, the label that is predicted is the same for the features before and after medication. These results reinforce the subjectivity adjacent to the MDS-UPDRS scale, a scale with low resolution, confirming the need to use the movement features as complementary information in the assessment of PD motor symptoms. These analyses emphasize how crucial it is to look closely at movement patterns and signals to comprehend how medication affects the symptoms completely. Doctors can monitor patients' progress and make more accurate, individualized therapy adjustments with the help of the vital information provided by the changes in movement characteristics seen before and after medication.

## VI. The iHandU System as the next PD Management Tool

The iHandU system's innovative combination of the prescription of both motor symptoms measure and medication intakes at well-defined schedules is a ground-breaking method of patient care in PD. A wide range of opportunities arises from the integration of the iHandU system as a patient management tool, especially when comprehending patient responses to therapy. When medication regimens and motor test data are combined, it is possible to thoroughly examine movement patterns both before and after therapy. This is seen in the case study of patient HPD05, which is used as an example that shows variations in movement characteristics across the day. Equipped with this kind of information and more precise data, such as the features, doctors have an enhanced decision-making capability to track patients' development better and tailor their care to meet each person's needs. It helps doctors to perform in-depth, individualized assessments of therapies that are delivered. The system's results are based on a punctual tests methodology rather than fully continuous monitoring as in many other systems. By ensuring that doctors receive exact, rigorous, and time-controlled quantitative outcomes, this feature improves the quality of information provided to them about the symptoms of their patients.

The clinical study performed was very helpful in giving a thorough understanding of the system outside of the hospital environment, which has allowed for important improvements. The System Usability Scale (SUS), a methodology incorporated in the questionnaires, was used to evaluate the system based on a wealth of feedback obtained, presented in Fig. 8. This scale quantifies the system's usefulness and provides a numerical depiction of users' opinions regarding its efficacy and ease of use. With an average SUS score of 83.9%, the iHandU system's usability rating is fairly good. This implies that the majority of users believe the system to be user-friendly and effective in fulfilling their needs. With such a high score, the system's functionality and design are highly in line with user expectations, resulting in a satisfying user experience. Furthermore, attaining an NPS score of 83.9%, which is categorized as a "promoter" on the scale, highlights the high degree of satisfaction that the 5 patients and 2 clinicians have with the system and their propensity to recommend it to others. This advocacy serves as a powerful dissemination tool that increases the system's potential effect and reach while also reflecting the positive experiences of the users.

Although the test population is small, since the comfort levels with technologies are distributed, this study confirms that the system might succeed as a future product and is well-suited to meet the demands of its intended users. Based on the results, it is possible to suggest that the system design and functionalities are validated, showing strong usability and high user satisfaction levels, which provide a basis for its widespread adoption and ongoing advancement.

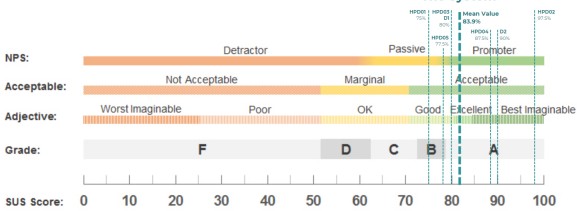

Fig. 8. Overview of the system Usability Scale (SUS) rating table for the iHandU System.

## VII. Conclusions and Future Work

The system has demonstrated significant evidences of success in monitoring PD motor symptoms outside of the hospital environment, showing patient management capabilities and having positive feedback from patients and doctors. Through the clinical study conducted at home, it was also possible to analyze whether variations in motor symptoms before and after medication provide relevant information about the patient's symptomatic profile, as complementary to the current MDS-UPDRS scale. These results suggest that it is an effective tool for remote and personalized monitoring of PD, improving the quality of information for doctors and potentially the therapy provided for patients. Additionally, prescribed monitoring, in a controlled continuous approach, can have higher gains that allow more reliable information to be obtained for a better and more personalized adjustment of therapy. Improvements according to feedback received from patients and doctors will be implemented, together with a deeper analysis and validation of the effect of medication on the expression of motor symptoms, contributing to the constant evolution of the system. To ensure the robustness of the implemented analysis and symptoms quantification, the system will continue to be tested in non-clinic scenarios, and more data will be collected outside of the medical checkup moment to strengthen the algorithms and improve their performance when applied at patients' homes.

## Acknowledgement

We thank the five patients who participated in this study.

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
