# OpenReview forum: "Creating the next Digital Telemedicine Tool for Parkinson's Disease Management with AI"
_IEEE.org/EMBS/BHI/2024/Conference — IEEE BHI'24_

### Official Review · Reviewer_baez · 2024-07-23
**Review of "Creating the next Digital Telemedicine Tool for Parkinson’s Disease Management with AI"**

**Overall Rating:** 7
**Confidence:** 5

**Review:**

Quality:
The paper is well-structured and presents a comprehensive approach to the development and testing of a digital telemedicine tool for Parkinson's Disease (PD) management. The use of AI and wearable technology is well-explained, and the integration of these technologies into a cohesive system is a notable strength. However, the study would benefit from a larger sample size and a more detailed discussion on handling class imbalance and data augmentation techniques.
Clarity:
The clarity of the paper is generally good, with a logical flow and clear presentation of the system components and study results. There are, however, several grammatical errors and inconsistencies in terminology that should be addressed to improve the overall readability.
Originality:
The work is highly original, addressing a significant gap in the remote monitoring and management of PD symptoms. The combination of real-time assessment using AI and wearable devices is innovative and has the potential to significantly enhance patient care outside of clinical settings.
Significance:
The significance of this work lies in its potential to improve the quality of life for PD patients by providing continuous, objective monitoring and personalized therapy management. The high usability score and positive feedback from initial testing indicate that the system meets user needs effectively.
Pros
Innovative Approach: Integration of AI and wearable technology for real-time assessment of PD symptoms.
Comprehensive System: Includes wearable device, mobile application, and web dashboard, providing a robust solution.
High Usability: Achieved a usability score of 83.9% (A grade) based on the System Usability Scale (SUS).
Real-World Testing: Tested both in clinical settings and at patients' homes, providing valuable data on effectiveness and usability.
Potential Impact: Significant potential to improve PD management and patient quality of life through continuous monitoring and personalized therapy.
Cons
Sample Size: The study included a relatively small number of patients (5), which may limit the generalizability of the results.
Class Imbalance: Significant class imbalance in the dataset could affect the accuracy of the machine learning models.
Data Augmentation: Lack of detailed discussion on the data augmentation techniques used and their effectiveness.
Subjectivity of MDS-UPDRS Scale: Reliance on the subjective MDS-UPDRS scale for symptom assessment.
Security Measures: Insufficient details on the security measures implemented to protect patient data.
Ethical Considerations: The paper does not explicitly mention the process of obtaining informed consent or the specific data protection practices employed.
Recommendations
Increase Sample Size: Future studies should include a larger sample size to improve the robustness of the findings.
Address Class Imbalance: Explore advanced techniques for handling class imbalance in the dataset.
Detailed Data Augmentation: Provide a more detailed discussion on data augmentation methods and their impact on model performance.
Mitigate Subjectivity: Investigate alternative or complementary methods to the MDS-UPDRS scale to reduce subjectivity.
Enhance Security Details: Include a thorough analysis of the security measures implemented to protect patient data.
Explicit Ethical Practices: Detail the process of obtaining informed consent and data protection practices to meet ethical standards.

**Other Quality Metrics:**

Clarity of Writing: Good
Clinical Significance: Great
Methodological Novelty: Excellent
Experiments and Results: Good

**Questions For The Authors:**

1. Are there plans to increase the sample size in future studies to enhance the robustness and generalizability of the findings?
2. How do you plan to address the significant class imbalance in your dataset in future iterations of the machine learning models?
3. Can you provide more details on the data augmentation techniques used in your study? How effective were these techniques in improving the model’s performance?
4. Given the known subjectivity issues with the MDS-UPDRS scale, have you considered alternative or complementary methods for symptom assessment? If so, what methods are you considering?
5. What specific security measures have you implemented to protect patient data? Can you detail the encryption methods, access controls, and any other data protection practices employed to ensure privacy and confidentiality?

**Strengths:**

1. The paper presents a unique combination of AI and wearable technology for real-time assessment of Parkinson's Disease (PD) motor symptoms, addressing a significant gap in patient monitoring outside clinical settings.
2. The system achieved a high usability score of 83.9% on the System Usability Scale (SUS), indicating a strong alignment with user needs and expectations.
3. The system includes a wearable device, a mobile application, and a web dashboard, providing a robust and integrated solution for continuous and personalized PD management.
4.The system has been tested in both clinical settings and at patients' homes, providing valuable data on its effectiveness and usability in diverse environments.
5. The system has significant potential to improve the quality of life for PD patients through continuous, objective monitoring and personalized therapy management, offering a more optimized approach to PD symptom management.

**Summary Of The Paper:**

This paper presents an innovative digital telemedicine tool for managing Parkinson's Disease (PD) using AI. The tool integrates a wearable device, a mobile application, and a web dashboard to provide real-time, objective quantification of PD motor symptoms in non-clinical settings. The system has been tested both in clinical environments and at patients' homes, demonstrating promising results with a usability score of 83.9% based on the System Usability Scale (SUS). While the study shows significant potential, improvements are needed in the areas of sample size, handling class imbalance, and providing detailed security measures. Additionally, explicit details on informed consent and data protection practices are essential. Overall, the system represents a valuable advancement in PD management.

**Weaknesses:**

1.There is significant class imbalance in the dataset, which can affect the accuracy and reliability of the machine learning models. More advanced techniques for handling class imbalance should be explored.
2. The paper lacks a detailed discussion on the data augmentation techniques used and their effectiveness in improving model performance. Providing more information on this aspect would strengthen the methodology.
3. The reliance on the MDS-UPDRS scale, which has known subjectivity issues, might limit the accuracy of the assessments. Investigating alternative or complementary methods to reduce subjectivity would be beneficial.
4. The paper does not provide sufficient details on the security measures implemented to protect patient data. A thorough analysis of data protection practices, including encryption and access control, is essential to ensure patient privacy and confidentiality.

---

### Official Review · Reviewer_19ps · 2024-08-12
**Creating the next Digital Telemedicine Tool for Parkinson's Disease Management with AI**

**Overall Rating:** 7
**Confidence:** 5

**Other Quality Metrics:**

(a) Clarity of writing: Excellent
(b) Clinical Significance: Excellent
(c) Methodological Novelty: Great
(d) Experiments and Results: Great

**Questions For The Authors:**

1. Future research should aim to increase the sample size to enhance the generalizability of the findings and provide a more robust validation of the system across diverse patient populations.
2. Incorporate more diverse datasets in the machine learning model training and validation process to ensure that the system is robust and applicable across various real-world scenarios.

**Strengths:**

The scope of this paper is highly relevant to the ongoing development and application of telemedicine and digital health technologies, particularly in the context of chronic neurodegenerative diseases like Parkinson's Disease. By enabling continuous monitoring, the proposed tool can generate a rich dataset that offers nuanced insights into symptom fluctuations and disease progression. This can also lead to more tailored and effective treatment plans. Moreover, the potential integration of this tool into existing healthcare infrastructures could streamline clinical workflows, enhance the efficiency of care delivery, and improve overall patient management. The importance of this scope is underscored by the increasing need for remote healthcare solutions, especially in light of recent global health challenges.

**Summary Of The Paper:**

The primary focus of the paper is the development and implementation of a digital telemedicine tool specifically designed for managing Parkinson's Disease using advanced artificial intelligence techniques. This innovative tool enables real-time monitoring and assessment of PD motor symptoms, allowing patients and healthcare providers to gain continuous insights into the disease's progression outside traditional clinical environments. By leveraging machine learning models, the system offers an objective, data-driven approach to evaluating symptoms, thereby facilitating more personalized and potentially more effective treatment strategies. Overall, the paper's language is formal and appropriate for an academic audience, with technical terms used accurately and effectively.

**Weaknesses:**

1. The study's small sample size, with only a limited number of participants, restricts the generalizability of the findings. The small sample size may not fully represent the diverse range of symptoms and disease progression in the broader Parkinson's disease population. This limitation needs to be addressed in future studies to validate the system's effectiveness across a broader population.
2. While the system shows promise in real-world settings, the reliance on data collected from a specific clinical environment may not fully capture the variability of PD symptoms across different patient populations.
3. There is an insufficient exploration of potential biases in data collection and data analysis. The lack of discussion on these biases could undermine the robustness of the study's conclusions.

---

### Decision · Program_Chairs · 2024-09-23

Accept